# Non-Invasive Continuous Optical Monitoring of Cerebral Blood Flow after Traumatic Brain Injury in Mice Using Fiber Camera-Based Speckle Contrast Optical Spectroscopy

**DOI:** 10.3390/brainsci13101365

**Published:** 2023-09-25

**Authors:** Dharminder S. Langri, Ulas Sunar

**Affiliations:** 1Department of Biomedical Engineering, Wright State University, Dayton, OH 45435, USA; langri.2@wright.edu; 2Department of Biomedical Engineering, Stony Brook University, New York, NY 11794, USA

**Keywords:** cerebral blood flow, speckle contrast optical spectroscopy (SCOS), laser speckle contrast imaging, traumatic brain injury

## Abstract

Neurocritical care focuses on monitoring cerebral blood flow (CBF) to prevent secondary brain injuries before damage becomes irreversible. Thus, there is a critical unmet need for continuous neuromonitoring methods to quantify CBF within the vulnerable cortex continuously and non-invasively. Animal models and imaging biomarkers can provide valuable insights into the mechanisms and kinetics of head injury, as well as insights for potential treatment strategies. For this purpose, we implemented an optical technique for continuous monitoring of blood flow changes after a closed head injury in a mouse model, which is based on laser speckle contrast imaging and a fiber camera-based approach. Our results indicate a significant decrease (~10%, *p*-value < 0.05) in blood flow within 30 min of a closed head injury. Furthermore, the low-frequency oscillation analysis also indicated much lower power in the trauma group compared to the control group. Overall, blood flow has the potential to be a biomarker for head injuries in the early phase of a trauma, and the system is useful for continuous monitoring with the potential for clinical translation.

## 1. Introduction

Traumatic brain injury (TBI) is a significant public health concern and is a leading cause of death and disability worldwide, especially in children, adolescents, and young adults [1]. Neurocritical care focuses on monitoring CBF to prevent or detect and treat secondary brain injuries before damage becomes irreversible [1,2]. Thus, there is a critical unmet need for continuous neuromonitoring methods to quantify CBF within a vulnerable cortex continuously and non-invasively. Existing neuromonitoring technologies have several limitations [1,2,3,4,5,6,7]. Invasive measures of regional CBF require surgical burr hole placement of a thermal diffusion or laser Doppler flowmetry probe, which attaches to a single, small location [3]. Transcranial Doppler (TCD) ultrasound (US) measures flow velocity within large cerebral arteries but cannot measure cerebral microcirculation and is challenging to measure over a long period of time [1].

The use of optical blood flow measurements and animal models for TBI and closed head injury (CHI) has been motivated by the need for a better understanding of the underlying mechanisms, biomarkers, and potential therapeutic targets for these conditions [8,9,10,11]. TBI can cause significant changes in cerebral blood flow and metabolism, which can lead to further damage and complications.

Commercially available optical methods such as Near-Infrared Spectroscopy (NIRS) use a diffuse optical technique that allows for non-invasive continuous monitoring of hemodynamics in vivo [6,7,12,13,14,15,16]. However, this technique only measures changes in the concentration of different chromophores and cannot directly calculate continuous CBF. Laser Doppler scanning and speckle contrast imaging have also shown promise for blood flow studies in animals but require extensive skull clearance and are limited in their use for bedside monitoring [17,18].

We used fiber camera-based speckle contrast optical spectroscopy (SCOS) [19,20,21,22,23,24,25,26,27,28,29] to measure blood flow in deep tissue. The standard technology for this is continuous wave diffuse correlation spectroscopy (DCS), which has limitations in terms of cost, speed, and the number of detectors that can be used. The fiber camera setup can incorporate multiple fibers in a single camera, making it more cost-effective and increasing both temporal and spatial resolution. In this research, we monitored blood flow changes in a TBI model in mice, which was caused by using a weight-drop apparatus, a well-established model for studying neurophysiological deficits and cell death. Accurate monitoring of cerebral blood flow (CBF) after head trauma is essential for injury prognosis and planning optimal treatment [13,30,31]. The research aimed to continuously monitor and quantify the changes in blood flow non-invasively by using a fiber camera-based approach using SCOS for measuring and monitoring CBF following closed head injury in a mouse model.

## 2. Materials and Methods

### 2.1. Study Design

The animal protocol was accepted by the department of Laboratory Animal Resources (LAR) of Wright State University. Twenty female C57BL/6J mice were used. The mice were anesthetized using an isoflurane vaporizer unit with 5% isoflurane while in the anesthesia chamber. After the initial anesthesia, the mice were given 2.5% isoflurane and fitted with a nose cone to maintain their unconscious state during the measurements.

The experiment was designed to investigate the changes in blood flow in mice after TBI induction, and the results were compared to the control measurements. The weight-drop technique was chosen for inducing TBI as it closely mimics focal head trauma, and the optical setup was used to record blood flow changes to study the effects of TBI on cerebral blood flow. The experiment involved a total of 20 mice, which were divided into two groups: 10 control mice and 10 mice designated for the induction of traumatic brain injury (TBI). The first step of the experiment involved taking control measurements in the control group of mice. These measurements were taken for a period of 30 min to establish a baseline measurement of blood flow. In the TBI mice group, the weight-drop technique was used to cause the TBI, which involved dropping a cylindrical metallic weight weighing 100 g from a height of 90 cm onto the head of the mouse between the anterior coronal suture and posterior coronal suture. After TBI was induced, the mice were immediately transferred to the optical setup platform to record optical measurements for a duration of 30 min. Throughout the process, the mice were kept under anesthesia to ensure that they did not experience any pain or discomfort. For the control and TBI groups, as soon as the 3 min optical measurements were complete, the mice were euthanized using the cervical dislocation technique.

### 2.2. Fiber-Based Speckle Contrast Optical Spectroscopy Setup and Analysis

The custom fiber-based SCMOS system is shown in Figure 1. It consists of a continuous-wave high-coherence laser source with a 785 nm wavelength (DL785-100-SO, CrystaLaser Inc., Reno, NV, USA), a sCMOS camera (Zyla 5.5 sCMOS, Andor, An Oxford Instruments Company, Concord, MA, USA), coupled with a zoom lens (MVL6X12Z, Navitar Inc., Rochester, NY USA) to adjust magnification and speckle size, a 600 μm core diameter multi-mode source fiber, and a multielement flexible custom imaging bundle from Schott with core diameter of 1.93 mm and 17,000 individual elements capable of collecting speckle patterns and transmitting them to the sCMOS camera. In general, the system has the capability to measure multiple source–detector combinations by scanning the source and by coupling multiple image fibers to the camera. However, here we used only one source–detector pair for the brain. To secure the source and detector fibers, a custom optical probe was produced using a 3D printer, and a flexible ninjaflex filament was used on each mouse’s head. The source and detector fiber separation was set at 5 mm, and the camera’s exposure time was set at 20 ms.

The images were acquired using MATLAB (MathWorks, Inc., Version: 9.10.0 (R2021a), Natick, MA, USA). The rCBF values were obtained through data processing and analysis. SCOS examines speckle patterns generated by the interference of coherent light scattered by particles, such as red blood cells. The movement of these particles causes changes in optical power and speckle pattern blurring, which can be used to estimate blood flow through speckle contrast $K_{s}$. The relationship between contrast and flow depends on the scattering framework, particle motion, and the presence of static scattering with the relationship of $1/K_{s}^{2}$ = BFI [19,20,21,22,23,24,25,26,27,28,29], where BFI is the blood flow index, which can be described as an estimation of perfusion, according to the speckle contrast optical spectroscopy theory. This method is easy to use and allows for real-time data processing, which is why it is widely used in many SCOS applications.

Speckle contrast, $K_{s}$, is given by $K_{s}=\frac{\sigma_{s}}{<I>}$, where <*I*> and $\sigma_{s}$ are the mean and standard deviation of the intensity in the surrounding area of a pixel. The acquired frames were corrected against the average intensity of 500 background images, where the intensity of the mean of dark background images was subtracted from the signal as dark noise correction before commencing data collection for the experiment, and an area of 200 × 200 pixels of the corrected speckle pattern was used for $1/K_{s}^{2}$ calculation.

In our initial experiment, we compared the SCOS results with a lab-standard diffuse correlation spectroscopy system, which was detailed in our previous publication [32]. Briefly, the DCS system consists of a long-coherence (~10 m) laser source (785 nm CrystaLaser Inc., Reno, NV, USA), eight NIR-optimized single-photon counting modules (SPCM-NIR, Excelitas, Vaudreuil-Dorion, QC, Canada), and an 8-channel auto-correlator board (Correlator.com), of which one channel was used. A single multi-mode fiber (200 μm core diameter, 0.39 numerical aperture (NA)) was used to guide the 785 nm laser light to the scalp, and a single-mode fiber (5 μm core diameter, NA of 0.13) collected the light emitted from the scalp to the single-photon-counting modules. The separation distance between the source and the detector fiber was set at 5 mm.

## 3. Results

In order to evaluate the custom fiber camera SCOS system, the results were compared to those obtained from the diffuse correlation spectroscopy (DCS) system, which is considered a standard in the field [9]. The DCS readings and the SCOS readings from the newly created device were taken at the same time during experiments involving both mouse TBI and control models, with each system using a single channel. The DCS system utilized a single-mode fiber as its detector fiber, and both detector fibers (single-mode fiber for DCS and imaging fiber for SCOS) were placed at the same distance from the source (5 mm) and as close to each other as possible. The results, depicted in Figure 2, showed that the fiber camera SCOS system was able to accurately detect and measure changes in blood flow and observe the expected trends, demonstrating its reliability for future in vivo studies on blood flow changes. The control model, where the mouse was anesthetized and in resting condition throughout the experiment, showed a relative blood flow change of less than 4% for both devices, as seen in Figure 2a, while in the TBI model experiment, the fiber camera SCOS system showed an approximately 15% change, while the DCS device showed an approximately 18% change 30 min after impact and the induction of TBI, as seen in Figure 2b. The trends of relative cerebral blood flow changes in both devices were closely correlated, validating the effectiveness of the developed device as compared to the established DCS technique.

After the device was validated through simultaneous comparison with DCS, mice experiments were performed to detect the capability of the device to differentiate between the TBI group and the control mice population based on the change in blood flow in the cerebral region.

The results showed that the mean relative blood flow change for the control mice was less than 4%, while the mean relative blood flow change for the TBI mice was around 10%, 30 min after impact (Figure 3a). Figure 3b indicates the average trace of percentage change in relative cerebral blood flow for both the control and the TBI mice model. The experimental results are presented as the mean (µ) and standard error (SE) of relative CBF of 10 sample mice in each case (μ ± SE) throughout the 30 min timeline of data measurement in the figure mentioned, where the standard error (SE) can be calculated by dividing the standard deviation (σ) of the sample by the square root of the sample size (n). This finding clearly demonstrates a significant difference between the control and the TBI population and that the technique can detect the difference between the two groups.

Further statistical analysis was carried out to compare the magnitude change in cerebral blood flow between the TBI and the control mice. The Wilcoxon rank sum test was performed, and the results showed a very strong and significant difference between the relative blood flow of the TBI mice and the control mice, with a *p*-value < $10^{-4}$. The results indicate that the device was sensitive in detecting the changes in cerebral blood flow due to TBI and has the potential to differentiate between the control group and the TBI group in mice.

Low-frequency oscillations (LFOs) can serve as a potentially novel metric for brain function, and optical blood flow and oxygenation can detect these oscillations [33,34,35,36,37,38,39,40,41,42,43,44,45,46,47,48,49,50,51,52,53,54,55,56,57]. To extract low-frequency oscillations (LFOs) from relative cerebral blood flow, the Welch method was used to obtain power spectral densities (PSDs), as was carried out in previous studies [58,59,60,61,62,63].

Figure 4 displays the mean and standard error of LFO spectrums for the 10 mice in each group, namely the control and the TBI population. A comparison of the results between the control and the TBI mice reveals that both groups have similarly shaped frequency responses. Notably, both groups display significant activity in the range of 0.01–0.06 Hz. Furthermore, the PSD of the TBI mice population is around 0.014 (a.u) at the very low frequency band of ~0.02 Hz, while for the control population, it is around 0.021 (a.u). These findings are consistent across all 10 mice in each group, as demonstrated by the standard error in Figure 4. Overall, the power spectra of spontaneous low-frequency oscillations show significant attenuation in all frequencies.

## 4. Discussion

This study shows that using optical blood flow measurements is a viable method for monitoring blood flow in mice with traumatic brain injuries. These measurements can provide continuous and non-invasive monitoring of blood flow, which can be critical in understanding the trauma stage, which can be helpful in neurocritical care as it aims to limit secondary injuries by closely monitoring the brain. The pilot study successfully measured blood flow in mice’s heads for approximately 30 min, demonstrating the feasibility of providing continuous, longitudinal measurements of blood flow. The results align well with those of Abookasis et al. [64], who observed a 13% decrease, and Witkowski et al. [65], who also observed a similar decrease by using the superficial optical speckle imaging method. Buckley et al. used DCS for monitoring repetitive concussions in mice for 24 h and several days and observed a cerebral blood flow (CBF) decrease at 4 h and a gradual daily based decrease in CBF until day 8 [66]. Fisher et al. observed an acute blood flow decrease using DCS within 30 min [67].

The attenuation of the power spectrum in the resting state or spontaneous oscillation in the TBI group are most likely due to the disruption of functional connectivity in brain networks due to induced trauma. Thus, low-frequency oscillations (LFOs) might be useful for providing feedback and identifying potential biomarkers for assessing the trauma stage and the efficacy of intervention.

It should be noted that there are potential limitations to our study. Although the head is a complex structure containing many tissue layers, these layers are very thin in mice (skull~500 μm). Therefore, here the head is considered a homogenous medium. The results obtained from the experiment represent the average values of a single volume. Another potential limitation of this approach is the need for space on the scalp for the contact optical probe, which may be occupied by numerous invasive and non-invasive other devices during multimodal measurements in future human use. To address this limitation, optical fibers can be arranged and placed as required in custom 3D-printed probes, which can reduce the additional space required. This can also enable routine correlation of optical metrics with other modalities for a more comprehensive understanding of the state of the monitored tissue. Other limitations include that the study only studied 30 min post-trauma; a longer time frame might be useful for mimicking clinical cases. The approach is not limited in this sense; it can allow continuous longitudinal monitoring for hours and days. Furthermore, the signal is averaged over a volume, and an imaging approach would allow for assessing focal traumatic changes [68].

## 5. Conclusions

In this study, we utilized a fiber camera-based speckle contrast optical spectroscopy (SCOS) technique to monitor blood flow changes in the brain following traumatic brain injury (TBI) in mice. This technique offers the advantage of being non-invasive and provides continuous measurements of cerebral blood flow (CBF), which is critical for understanding the pathophysiology of the brain following trauma.

SCOS has the potential to become a valuable tool for monitoring brain hemodynamics in a clinical setting, such as in the NICU or in the field. This is particularly important as monitoring CBF changes in real-time following TBI can help guide intervention and prevent secondary brain injury. The non-invasive and continuous nature of SCOS can provide clinicians with the necessary information to make informed decisions in real time, which could have significant impacts on patient outcomes.

## Figures and Tables

**Figure 1 brainsci-13-01365-f001:**
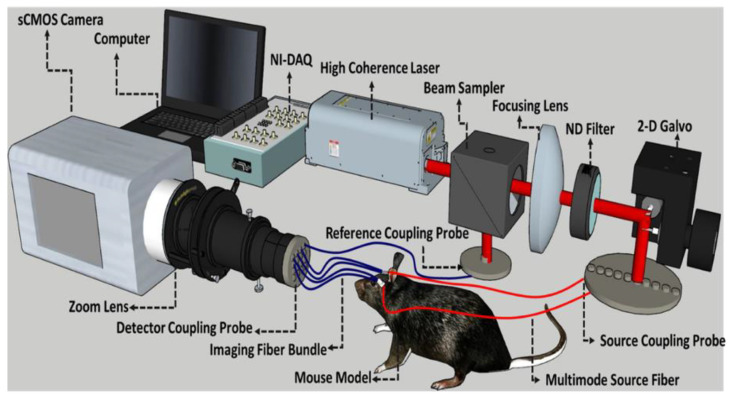
Block diagram of the SCOS device setup illustrating various components and their interconnections.

**Figure 2 brainsci-13-01365-f002:**
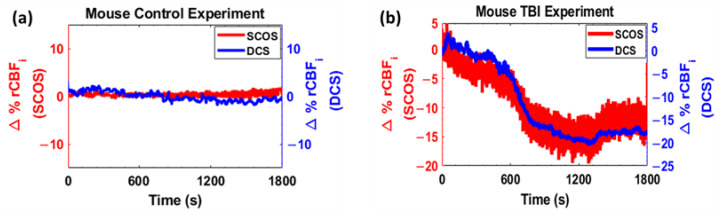
(**a**) Plot of change in CBF for control mice over 30 min using DCS and SCOS. (**b**) Plot of change in CBF for TBI mice for 30 min after impact using DCS SCOS.

**Figure 3 brainsci-13-01365-f003:**
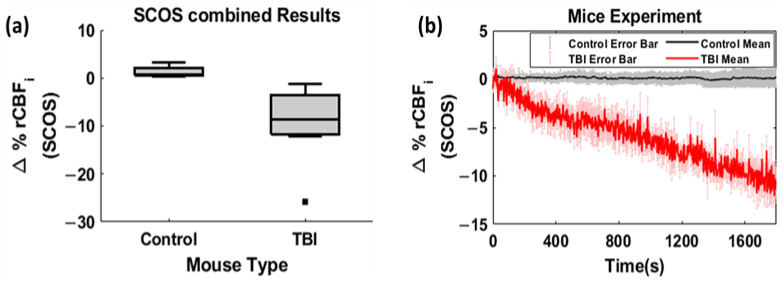
(**a**) Relative cerebral blood flow of control and TBI mice groups. (**b**) Time kinetics with mean and standard error bars of the trace of change in CBF for 10 control and 10 TBI mice each.

**Figure 4 brainsci-13-01365-f004:**
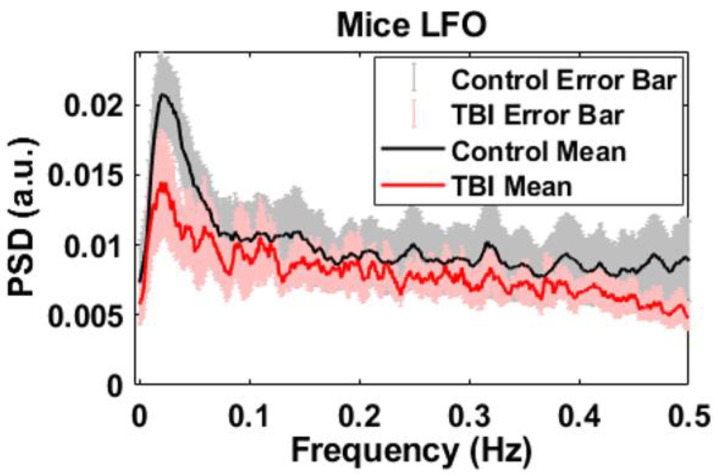
Mean and standard error of frequency spectrum of CBF for control and TBI.

## Data Availability

Data may be available via directly contacting via email request.

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
