# Peer review of "Non-Invasive Continuous Optical Monitoring of Cerebral Blood Flow after Traumatic Brain Injury in Mice Using Fiber Camera-Based Speckle Contrast Optical Spectroscopy"

_brainsci, 2023, doi:10.3390/brainsci13101365_

Round 1

Reviewer 1 Report

This work presents the development of a known technique (speckle contrast optical spectroscopy, SCOS) and its application to study a traumatic brain injury (TBI) model in mice. The authors present data comparing the new device with a similar standard technique (Diffuse Correlation Spectroscopy, DCS) and their results before and after TBI in mice with SCOS. Although the results are original, the manuscript is confusing and poorly written. It does not present all the relevant information in any section and introduces new methodological information during the results. Overall, there is no apparent reason that justifies the publication of the manuscript at this moment. Here are some drawbacks that I found and that I think should be addressed before considering publication: 

1. The introduction (and the work) lacks a clear goal and context. The authors do not justify the innovation and/or relevance of this work. Compared to other traditional SCOS devices, is there any novelty in instrumentation design? If so, which one? Has SCOS been employed in TBI before? If not, why would SCOS be better than DCS or NIRS? Even if there is no novelty in instrumentation or the application, the context must be better addressed in the introduction, which currently is too short and does not provide all context. 

2. The methods are also very unclear. The authors state that "the experiment involved a total of 20 mice (...) divided into [two] groups" (l. 94-95). However, based on the remaining text (and the results), it is inferred that the same mice underwent 30 minutes of baseline and then 30 minutes after the induction of TBI following baseline. In this case, where do the two groups come from? Were used a total of 10 mice in two different moments? Or 20 mice, 10 for control and 10 for TBI? If the latter, why were the 10 baseline mice euthanized? Please, clarify the experimental protocol (both materials and methods) and present the results accordingly. 

3. The use of the DCS device in the first result is surprising since this was not mentioned in the protocol. We were never informed which DCS device was used, how many sources and detectors, or the position relative to the LSCI probe. Also, it’s unclear why the DCS device was used for only one mouse (as mentioned in l. 193-194) and not all mice.

4. On the same topic, the authors claim that during the TBI experiment, there were 15% and 18% changes in the fiber-camera LSCI and DCS devices, respectively (l. 181). However, it’s unclear when the changes were measured during the 30 min period and the uncertainty of these changes. 

5. Although the authors correctly infer that "the trends of relative cerebral blood flow changes in both devices were closely correlated" (l. 182), the results presented in Figure 2 clearly show that SCOS data are noisier than DCS. In this case, why would one use SCOS instead of DCS? What are the sources of noise? And can the noise be improved? If one of the goals is to characterize the device built, then these questions should be discussed. 

6. The authors state that “the device was validated in the tissue mimicking dynamic phantom” (l. 193), but it does not provide any information about such validation. 

7. The authors state, "The experimental results are presented as mean and standard error of relative CBF” (l. 202-203), although it’s unclear where these results are. 

8. The authors claim that “the results showed a very strong and significant difference between the relative blood flow of TBI and control mice.” (l. 211-212). Wouldn’t that be a logical and expected result?

9. In addition, the claim that “the results indicated that the device was sensitive in detecting the changes in cerebral blood flow due to TBI and had the potential to differentiate between control patients and TBI patients” (l. 212-214) does not seem to hold based on the experiment and the results presented in this work. Both SCOS and DCS measure relative changes compared to a baseline from the same subject. In order to differentiate between TBI patients and controls, it’s necessary to make absolute comparisons. Please clarify. 

10. The references for low-frequency oscillations are biased and do not offer a comprehensive overview of LFOs in the field of diffuse optics. In addition, LFOs are defined in the ~0.01-0.1 Hz; therefore, having peaks within this range would not necessarily mean a significant difference that can be attributed to TBI only. If the authors want to compare the LFOs between the two conditions, the contributions across the whole LFO spectrum should be more appropriate. 

11. In the same vein, it’s not clear how the authors can argue that “(LFOs) might be useful feedback and (a) potential biomarker for assessing the trauma stage and the efficacy of intervention” based on the results presented in Figure 4 and the last paragraph of the results. 

There are a few typos throughout the manuscript that should be addressed. Overall, the quality of English is good. 

Author Response

please see the attachment, thank you.

Reviewer 2 Report

This manuscript deals with blood flow monitoring of mouse brain using diffuse optical methods based on speckle contrast measurement. Although the animal experiment seems to be done carefully enough with enough number of mice, the way it is presented in manuscript has not quite reached the publishable quality. A revision based on the following comments will make a much better manuscript with scientific rigor.

[Experimental parameters]

It is hard to get the detailed picture about how the experiment has been carried out. Although the whole setup is shown in Fig 1, it could be misleading because the Galvo and multi-channel fibers do not seem to have been used in real experiment, according to the text. On the other hand, important information such as the details of detector fiber (NA, number of cores, etc), pixel resolution of sCMOS, exact position of source and detector fiber on the mouse brain, are all missing. The fiber core diameter, especially, has to be compared with respect to both speckle size on the brain and pixel size of sCMOS, in order for the speckle-contrast-based blood flowmetry to work.

[Analysis]

It is interesting that TBI group consistently shows less LFO level than control group does. But what is observed in power spectrum should be also explainable in time-domain as well. Did the authors observe the ~0.02 Hz oscillation in time series data of BFI? It is strongly suggested that a time-series graph of either raw data or band-pass filtered data should be plotted, to see if the oscillation pattern around 0.02 Hz frequency is visible.

<Minor Issues>

-         Exposure time of 20ms sounds a bit too long. Normally 1-3ms is being used for physiological microcirculation measurement. Any reason to choose 20ms?

-         LSCI is a term for shallow blood flow measurement technique, where single-scattering event is dominant. Better not use interchangeably with SCOS. In the legends, as well, ‘sCMOS+LSCI’ need to be changed into ‘SCOS’, in order to avoid confusion.

-         200 x 200 pixels seem to be too big for calculating a single K, and it is doubted that light intensity was pretty much uniform across this big area. Better divide the ROI into several smaller areas and get the distribution of K’s, so that authors can calculate the mean and std of BFI.

-         line 179, 199 : ‘approximately 0%’ is a statement far from being scientifically rigorous, as it sounds like authors are claiming the error (or change) is negligible. Having a small error is good, but you need to be able to estimate how small the error is. Use ‘changes were less than P%’ instead, where P would be whatever number you confidently think is correct, be it 0.1 or 0.01.

-         What is the time resolution of DCS and SCOS, respectively? Why is SCOS data showing much bigger fluctuation in Fig2-b?

-         Chapter 3 is missing, which seems to be a simple numbering error. 

-         Fig 4 : Show how the calculated PSD can have the units of mm^2/s/Hz. In the first place, the units of BFI are different between DCS (mm^2/s) and SCOS (unitless), and PSD calculation requires squaring the Fourier transformed variable, so it is doubted that PSD’s unit is what it is in the paper.

-         line 214 : This experiment is dealing with mice, not ‘patients’.

-         Fig 3(a) : What does the title ‘Camera Results’ mean? Better change to more specific title.

Author Response

please see the attachment, thank you.

Round 2

Reviewer 2 Report

Great revision with clear rebuttals. Green light for publishing.